# Accelerating Attention Based Models via HW-SW Co-Design using Fine-Grained Sparsification.

Abhimanyu Rajeshkumar Bambhaniya*, Amir Yazdanbakhsh†, Suvinay Subramanian‡, Tushar Krishna*

*Georgia Institute of Technology. *abambhaniya3@gatech.edu,tushar@ece.gatech.edu*

†Google DeepMind. *ayazdan@google.com*

‡Google. *suvinay@google.com*

*Abstract*—This paper proposes FIne-Grained Sparsification (FIGS), a novel architecture for accelerating attention-based models using N:M structured sparsity. Existing hardware accelerators focus on optimizing compute to achieve ideal processing element (PE) utilization but ignore the implications of higher input bandwidth. FIGS overcomes this challenge by leveraging techniques like grouping and reusing input data to reduce required input bandwidth, achieving high PE utilization while minimizing on-chip interconnect area. The paper also proposes FIGS-Train, a sparsity training recipe that improves the accuracy of N:M structured sparse attention models.

## I. INTRODUCTION

Attention-based models have become increasingly popular in recent years due to their ability to focus on relevant information while processing input data. They are particularly effective in natural language processing (NLP) [3], [7], [10], image recognition [25], and speech recognition, code generation [4]. The attention mechanism allows these models to assign varying levels of importance to different parts of the input data, resulting in improved accuracy and efficiency.

Despite their effectiveness, attention-based models have become increasingly complex and computationally demanding, with some models containing billions of parameters [23]. This growing size of the models has led to longer training and inference times, limiting their applicability in practical settings. Various techniques, such as parallel processing, model compression, and low-precision arithmetic, have been proposed to overcome these challenges. However, these techniques have their limitations and often come at the cost of reduced accuracy [2], [5], [11]–[13], [18], [21], [22], [24], [28], [29].

Sparsity is an increasingly prevalent technique for accelerating attention-based models [6], [26] . Sparsity aims to reduce the number of parameters and computations required by the model by identifying and removing redundant or less important connections. N:M structured sparsity is a particularly interesting technique as it removes a fixed number of weights for each block of weights in the model [29]. This approach is more hardware-friendly and easier to implement in hardware accelerators than other sparsification techniques, such as random or magnitude-based pruning. By using this technique, it is possible to accelerate the execution of attention-based models while maintaining high accuracy levels [1], [20], [22].

Although N:M sparsity has been shown to be an effective technique for accelerating attention-based models, the existing

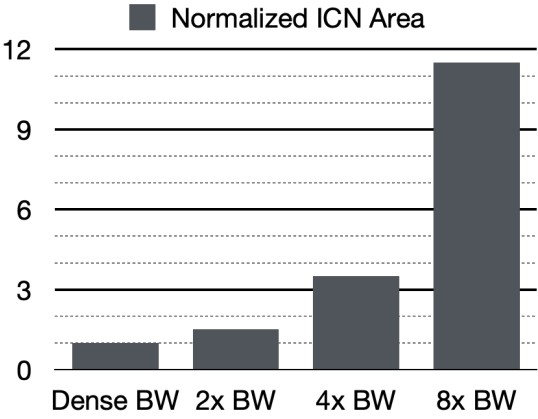

Fig. 1. **Growing cost of supporting higher on-chip bandwidth in 7nm.**

hardware accelerators, such as S2TA [21], STA [20], and VEGETA [15], focus primarily on optimizing the compute to achieve ideal processing element (PE) utilization, while ignoring the implications of higher input bandwidth. Recent research [27] has shown that the required input bandwidth for N:M accelerators scales up with M, which can lead to significant challenges in designing hardware that can effectively support this higher bandwidth. Figure 1 shows the increase in the on-chip interconnect area as the bandwidth required increases [16], [17].

Thus we propose FIGS: FIne-Grained Sparsification, novel architectures that can improve PE efficiency without increasing the on-chip bandwidth requirement. Our architectures leverage techniques like grouping and reusing input data to reduce the required input bandwidth. By doing so, they can achieve high PE utilization while minimizing the on-chip interconnect area. We also, show a potentially promising training approach, FIGS-Train, that helps improve the accuracy of an N:M structured sparse attention model. We develop even more efficient accelerators for attention-based workloads by combining fine-grained sparsity with FIGS hardware architectures designed to improve PE efficiency.

To summarize, we make the following contributions:

- We propose a new hardware microarchitecture, FIGS, that helps accelerate N:M structured sparse models without increased input BW requirements.
- We propose a new sparsity training recipe, FIGS-Train,

that can be accelerated using N:M structured HW.
- We compare FIGS microarchitecture, with current SOTA N:M accelerators.
- We train attention-based models using FIGS-Train, achieving better accuracy than current state-of-the-art N:M structured sparsification techniques.

## II. FIGS ARCHITECTURE

### A. Processing Elements(MAC PE)

Each processing element comprises of $\beta$ multiplier units. The FIGS architecture works in a systolic array-inspired weight-stationary format, storing all sparse weights in a MAC PE register and metadata. We have bitmask as metadata, as that would result in the lowest overhead for the structured sparse case. The MAC PEs send out $\beta$ metadata to its corresponding *Swap Reg*, and it gets the appropriate input activation input. Each multiplier and adder completes the MAC operation and forwards an output to the downstream MAC PE. We position $\beta$ as a configurable parameter, as with newer technology nodes, it is possible to have fewer registers between PEs [14], [17]. This helps in reducing the area of the complete engine.

### B. Swap Reg

*Swap Reg* acts as input staging unit. It takes in $\gamma$ input words from the engine input or the previous swap reg. Each *Swap Reg* registers these $\gamma$ input words in flops. The *Swap Reg*'s main task is to provide appropriate data to the MAC PEs. Using the meta-data provided by the MAC PEs, the *Swap Reg* would generate $\beta$ corresponding output words. Multiple neighboring *Swap Regs* can also talk to each other and exchange their stored input words. Depending on the architectural parameter,$\alpha$, a single *Swap Reg* takes all the input activation of the $\alpha$ *Swap Regs*, making a total of $\alpha \cdot \gamma$ input words. This means each *Swap Regs* has to select the $\beta$ output activation words from $(\alpha + 1)\gamma$ inputs. Thus a bigger $\alpha/\beta/\gamma$ value results in a bigger *Swap Reg*.

### C. The FIGS Engine

We build the complete FIGS engine using MAC PEs and *Swap Reg* as building blocks. A FIGS engine has $N_{Rows} = R$ and $N_{Cols} = C$. All the MAC PEs are accompanied by a *Swap Reg* at their input. Thus the whole FIGS engine has R * C MAC PEs and *Swap Regs*.

$\gamma$ depends on the maximum allowable on-chip bandwidth capacity and required sparsity support. $\beta$ will depend on the engine's required running frequency and the synthesis technology node. Theoretically, this can be any value between 1 and R. Finally, $\alpha$ depends on the maximum sparsity support required. We calculate $\alpha = \frac{M_{max}}{N_{min} \cdot \gamma}$.

Once, we have all the configuration parameters of the FIGS engine, we get an engine with the total number of MAC units $= R * C * \beta$. The engine takes in $C * \gamma$ words as input per cycle and generates $R * \beta$ output words per cycle.

### D. Dataflow

With the full FIGS engine now described, we move on to how the FIGS accelerates flexible N:M sparsity without changing the input BW. In Figure 3, we show 2:4 sparse matrix with input BW twice of the dense architecture. Each row of A matrix with 2:4 structured sparsity is mapped to a single column of MAC PEs. Each column of MAC PEs takes $\beta$ of rows of matrix A and generates $\beta$ partial sum. To support 2:4 sparsification, with have need $\alpha = \frac{M}{N \cdot \gamma} = 1$.

In the current configuration, two neighboring *Swap Regs* can exchange the input activations. We can see this in action, when in row 1 of the engine, the First MAC PE, takes Element 3 as input based on the metadata. This element has been provided by the *Swap Reg* of row 2. In each cycle, the *Swap Reg* forwards $\gamma = 2$ input activation to the next *Swap Reg* but gives out $\beta = 4$ activations to the MAC PE. Using this mapping, we ensure, that PEs are fully utilized while allowing higher $N{:}M$ sparsity, for the same $\gamma$. The full execution of this would take 'T' cycles.

Now, in order to execute 1:2 or 1:1 sparsity, we can still use the same mapping; just the *Swap Regs* would not need to swap data at that time.

## III. FIGS-TRAIN

### A. The Recipe

Now that we understand the abilities of the FIGS accelerator, we propose FIGS-Train, a specialized training recipe capable of efficient sparsification. Figure 4 shows the training schedule for FIGS-Train. The intuition is that by keeping the same number of non-zeros in the row, we keep reducing the block size. It is important to understand each progressive step would be a subset of the previous sparsification. For example, in the figure, we 1:4 $\in$ 2:8 $\in$ 4:16 $\in$ 8:32. Thus, this approach helps the gradients gather locally before pruning which helps achieve better accuracy.

### B. Training methodology and results

We applied this technique for weight sparsification at various locations in two attention-based models(ViT [9] and Swin V2 [19]. We trained these models on imagenet-1k [8] dataset with various amounts of sparsity. Table I summarizes the results for the 2 models at different sparsity levels. We compare the results with SR-STE [29], the current SOTA technique of N:M structured sparsification.

We found the two feed-forward layers in each layer are the most robust. Hence we always sparsified the weights in those layers. Next, we also pruned the weights of Query and Key matrices. We found pruning the weights of the Value matrix had the biggest impact on the network accuracy; Hence we did not prune the Value layer's weights.

## IV. EVALUATIONS

### A. Experimental Setup

We convert the feed-forward layer of the encoder block of 2 attention-based models to GEMM operations. We analyze

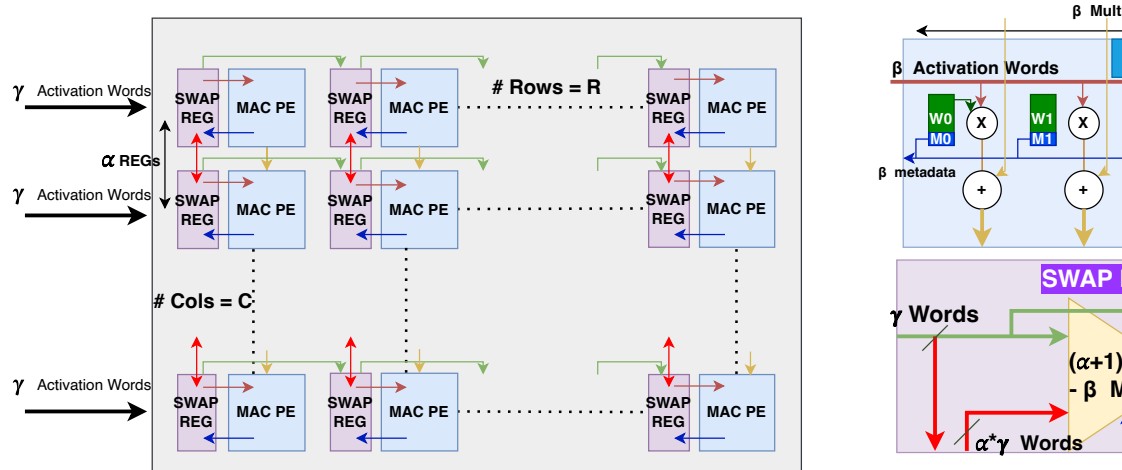

Fig. 2. **FIGS Micro architecture design for** $\beta = 4$ **and** $\alpha = 2$.

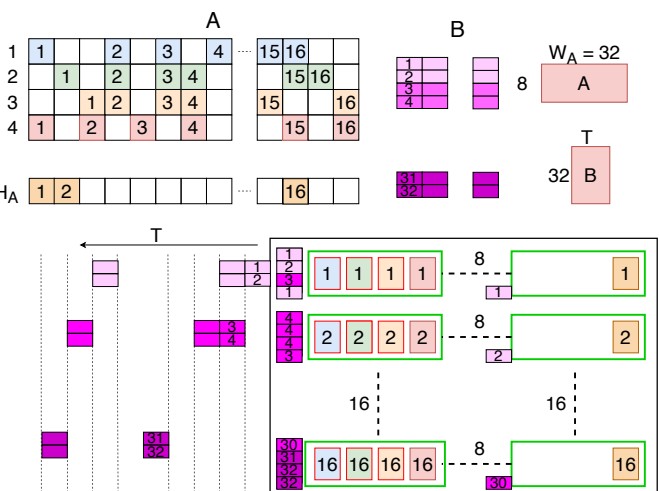

Fig. 3. **Dataflow for runing 2:4 structured sparse matrix multiplication with #Cols = 16, #Rows = 8,** $\gamma = 2$, $\beta = 4$ **and** $\alpha = 1$. **Note this other design SOTA design requires a minimum** $\gamma = 4$.

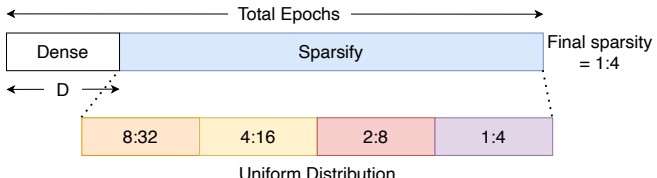

Fig. 4. **Breakdown of training epochs for FIGS-train recipe.**

TABLE I
SPARSITY TRAINING RESULTS WITH FIGS. FF MEANS SPARSITY IS PRESENT IN THE FEED-FORWARD LAYERS. QK MEANS THE SPARSITY IS PRESENT IN THE QUERY AND KEY WEIGHTS.

| Model | Sparsity | ViT | Swin V2 |
|---|---|---|---|
| Dense | / | 76.369 | 83.45 |
| SR-STE | 1:8(FF) | 77.869 | 81.437 |
| FIGS | 1:8(FF) | **78.175** | **81.466** |
| SR-STE | 1:8(FF + QK) | | 81.218 |
| FIGS | 1:8(FF + QK) | | **81.438** |

### B. Performance Analysis

Figure 5 shows the runtime of the SWIN and VIT Feed-Forward layers on the 5 accelerators. We normalized the runtime using the longest runtime (runtime for a dense systolic accelerator). We first observe that a systolic accelerator with no acceleration ability for sparse operations results in the highest runtime. STA/VEGETA is a state-of-the-art architecture for accelerating N:M structured sparsity, but these are bottlenecked by the input bandwidth to the compute array. We assume 4x the dense input bandwidth for all architectures. With this, STA/VEGETA can only accelerate layers with 1:4 or higher amount of sparsity. Thus for ViT, it can accelerate only the last sparsification phase with 1:4 sparsity, while for SWIN, it can accelerate the last 2 sparsification phases with 2:4 and 1:2 sparsity. Compared to these, FIGS architectures with the same input BW perform much better. FIGS($\alpha = 1$) can accelerate the sparsity ratio upto 2:8, FIGS($\alpha = 2$) can accelerate the sparsity ratio upto 4:16, and FIGS($\alpha = 3$) can accelerate the sparsity ratio upto 8:32.

Hence, we observe that all FIGS configurations perform much better than existing SOTA accelerators. For ViT with 1:4 sparsity, FIGS with ($\alpha = 3$) achieve 4.4x speedup over S2TA and Vegata, while 2.41x speedup for Swin v2 with 1:2 sparsity.

the training time for the layer when trained with the FIGS-Train recipe. We compare the runtime of these operations on a systolic array, SOTA N:M accelerators like STA/Vegeta, and compare them with three configurations of FIGS. We assume the same number of MACS = 512 MACs for all architectures.

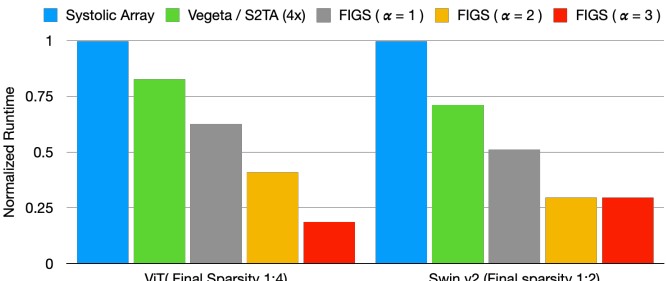

Fig. 5. **Runtime Comparison of fine-grained runtime sparsification.**

## V. FUTURE WORK

Some of the potential directions we are considering for this work are:-

- Our proposed FIGS technique provides a promising direction for accelerating attention-based models with N:M structured sparsity. One possible direction for future work is to investigate the efficacy of FIGS-Train in larger models and models other than those for image classification.
- We would also explore how to club this technique with other existing sparsification techniques like block sparsity, butterfly sparsity, etc.
- We would also like to do an area-energy-performance analysis of FIGS uarch to cap its ability and view its feasibility in realistic implementations.

## VI. CONCLUSION

In this work, we proposed FIne-Grained Sparsification (FIGS) techniques to accelerate attention-based models without increasing the on-chip bandwidth requirement. We showed that our proposed FIGS microarchitecture can achieve high PE utilization while minimizing the on-chip interconnect area. Our proposed FIGS-Train training recipe also showed promising results in improving the accuracy of N:M structured sparse attention models. We compared the performance of our proposed FIGS microarchitecture with state-of-the-art N:M accelerators and showed that it outperforms existing approaches.

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
