# OpenReview forum: "Accelerating Attention Based Models via HW-SW Co-Design using Fine-Grained Sparsification"
_iscaconf.org/ISCA/2023/Workshop/ASSYST — ASSYST Oral_

### Official Review · Reviewer_nC9R · 2023-05-05
**This work presents an accelerator that leverages N:M sparsity to develop an accelerator for DNNs.  The accelerator is tested on multiple modern networks and compared against other accelerators.**

**Rating:** 5
**Confidence:** 4

**Review:**

This work presents an accelerator and training technique that leverages N:M sparsity to improve the execution of modern DNNs.  The overall quality of this work is okay although there is room for improvement.  In particular, as a reviewer I had issues with the clarity of the writing and understanding what the authors are claiming in detail.  Given the short nature of the submissions, some of this is understandable however, I found things like Figure 3 being too busy and confusing due to items such as undefined variable names (Ha for example).  This forced me to read parts of the submission multiple times to ensure I gave the authors proper due.  Thus, I would like to see some refinement on the flow and writing to improve this work.

As for the originality, the work seems very similar to some of the works it compares against at a high level. However, there is some differences in the details, and I would like to see this brought out more explicitly.  I do appreciate the addition of the training section and the comparison against another training technique.

There is one issue with this work that seems to hold it back for me.  The authors make a claim that the bandwidth scales with the M parameter and cite the Sparseloop work of Wu et al. for this.  In that work, they are talking about the fact that to fetch sparse data that is stored in dense format and get a speedup you need to fetch based on the dense grouping and the speedup you are targeting. The authors do not seem to explain how FIGS bypasses this bandwidth increase.  In fact, I am not sure what exact format is stored in memory.  Without a gather unit the sparse data would be dense, and you still need an M factor bandwidth to fetch at a certain rate.  While I am certain the authors have looked into this, the submission fails to describe it in enough detail that this reviewer understands how FIGS improves this bandwidth.  Given the issues I had with the writing, it could be that I just misunderstood.  This is a large concern for me with this work.

**Review (Strengths/Weaknesses):**

#Strengths
This is an interesting topic that has a lot of potential.
The workloads chosen are good choices.
The addition of a training recipe is good.
#Weaknesses
The argument for bandwidth savings not clear.
It is not clear how data is stored in memory.
The writing needs to be improved.
Some of the figures are confusing.

**Reviewer Expertise:**

Knowledgeable: I used to work in this area and/or I try to keep up with the literature but might not know the latest developments.

---

### Official Review · Reviewer_Aup4 · 2023-05-05

**Rating:** 5
**Confidence:** 5

**Review:**

This paper proposes a co-design for accelerating structured sparsity in attention-based models. When using N:M structured sparsity, the bandwidth for loading inputs is increased. The proposed architecture features the design of Swap Reg to fetch inputs from neighboring Swap Regs to feed to MACs. The training recipe introduces a progressive structural sparsification method.

**Review (Strengths/Weaknesses):**

Strengths:
- The higher on-chip bandwidth for inputs under N:M sparsity is a well-motivated problem.
- The training recipe is intuitive and seemingly effective.

Weaknesses:
- The proposed architecture with the design of swap regs is less convincing without quantitative comparisons on hardware costs, e.g., area overhead and performance/area. The contribution is unclear without comparing with the baselines that simply increase on-chip bandwidth, in terms of area efficiency. The swap regs take non-negligible on-chip area, and the control and dataflow design are nontrivial.

**Reviewer Expertise:**

Expert: I have written one or more papers on this topic and/or I currently work in this area.

---

### Official Review · Reviewer_3tyh · 2023-05-05
**ML training accelerator accelerator for N:M fine-grained sparsity, combined with SW optimization techniques for attention. Tackles an interesting problem, but the paper could benefit from more clarity and focus.**

**Rating:** 4
**Confidence:** 3

**Review:**

## Summary of paper:
This paper proposes a HW/SW codesign for accelerating ML training (specifically focusing on attention), leveraging N:M structured sparsity. The proposed hardware uses a systolic array-like microarchitecture, consisting of a set of parameterized multiply and accumulate processing elements coupled to swap registers used to communicate inputs/activations with adjacent PEs. The engine can be configured based on available on-chip bandwidth and required sparsity support. The authors propose a training recipe to progressively modify sparsity to enable more efficient training while achieving higher accuracy. The authors evaluate their training recipes against SR-STE, and evaluate runtime of their hardware against N:M accelerators.

*Explicit strengths and weaknesses listed in the next section, after detailed comments.*


## Detailed Comments

Thank you for submitting to ASSYST! I liked the motivation behind your work. The introduction crisply summarized why we need to address the IO problem in accelerators, especially as we are increasingly reliant on sparsity in current accelerators (e.g., the A100 sparse tensor core). I specifically liked the focus on on-chip interconnects, and I think Figure 1 motivates the problem well. I believe a well-executed hardware/software co-design targeting this issue will be impactful.

The presentation of the hardware and software mechanisms proposed in this work unfortunately make it difficult to understand its core ideas and contributions.
- Hardware
    - The presented architecture, at a high level, makes sense. However, it is difficult for me to see where your innovations are. For example, you mention that a key technique you use is “grouping and reusing input data to reduce the required input bandwidth”. Do you get this benefit just by using a systolic array, or do you have some new insight here? At the surface, it looks like you combine a systolic array with hardware structures for (e.g., muxes) fine-grained sparsity, but how this connects to the goal fails to come out.
    - I think a big reason for this is that the text in II could significantly benefit from additional clarity. For example, it is difficult to understand exactly what each parameter (alpha, beta, gamma) means. Furthermore, some insight in how adjusting each parameter impacts cost/sparsity/performance would help.  Perhaps a table would help here?
    - There are numerous places where vague terms are used. For example, you say “Mac PEs send out \beta metadata”. What is the metadata?
    - It is not easy to understand how N:M fine-grained sparsity actually maps to your microarchitecture. The Dataflow section could benefit from clarity. For example, it is quite difficult to understand what Figure 3 means, and how that maps to your hardware parameters.

- Software
    - The presented software / training algorithm needs significantly more details. A paragraph is too little information to understand the key insight of your algorithm. You still have ample room to address this in the paper (1 and a half columns until the soft page limit).
    - Similarly, for your training evaluation, more details are warranted. For example, Table 1 reports “results” for your experiments. What is the metric? What is the validation set? What does FF and FF + QK mean?

- Hardware Evaluation
    - Same point here about a lack of methodology. There’s just simply insufficient understanding of how these experiments were performed to be able to take the results with more than a grain of salt. For example, you compare to STA (S2TA? - you say STA in the text and S2TA in the paper) / Vegeta. How is this done? In hardware? Emulation? Furthermore, it is unclear what a “Systolic Array” baseline is. There is no information about the microarchitecture, the text simply says “a systolic array”.
    - You report normalized runtime, but there is no evaluation of any other metric such as area/power, which makes Figure 5 potentially misleading. *This is especially critical given your claim in the introduction that your work address on-chip network area overheads.*

Finally, it is unclear to me the overall applicability of your work to attention. Sparsity can be applied to much more beyond attention. Other than mentioning the word “Attention” in the title, and using attention-based models in your evaluation, how does your work relate to attention specifically?



**Review (Strengths/Weaknesses):**

Strengths
- The paper lays out an important and timely problem. Specifically, it motivates well the need for techniques to address the IO-requirements of hardware, especially as techniques such as N:M sparsity increase on-chip network demand.
- The parameterized design of the microarchitecture enables an interesting trade-off analysis on training performance, accuracy, and hardware cost.
- The evaluation results show significant performance benefits to current hardware.


Weaknesses
- The paper would significantly benefit from additional clarity. The hardware is presented in a way that makes it difficult to understand its core insights and how/why it addresses its goal of minimizing on-chip interconnect area. The proposed software co-design is very limited, and largely lacks connection to the hardware.
- It is unclear why the HW/SW specifically benefits attention models.
- The evaluation lacks sufficient discussion of methodology. It does not evaluate the stated goal of reducing interconnect area.


**Reviewer Expertise:**

Little or no familiarity.